Projection of premature mortality from noncommunicable diseases for 2025: a model based study from Hunan Province, China, 1990–2016

Xu Qiaohua 1
Zhou Maigeng 2
Jin Donghui 1
Zeng Xinying 2
Qi Jinlei 2
Yin Li 1
Liu Yuan 1
Yin Lei 1
Huang Yuelong hylong410@126.com 1
1 Department of NCDs Control and Prevention, Hunan Provincial Center for Disease Control and Prevention , Changsha , Hunan , China
2 National Center for Chronic and Non-communicable Disease Control and Prevention, Chinese Center for Disease Control and Prevention , Beijing , China
Orlov Yuriy
Electronic publication date: 2020 Nov 3
Publication date: 2020
Volume: 8
Electronic Location ID: e10298
Received 2020 Jul 13; Accepted 2020 Oct 13
Copyright: ©2020 Xu et al.
Copyright year: 2020
Copyright holder: Xu et al.
License: This is an open access article distributed under the terms of the Creative Commons Attribution License, which permits unrestricted use, distribution, reproduction and adaptation in any medium and for any purpose provided that it is properly attributed. For attribution, the original author(s), title, publication source (PeerJ) and either DOI or URL of the article must be cited.
License URL: https://creativecommons.org/licenses/by/4.0/

Keywords: Projection, Noncommunicable diseases, Premature mortality, Joinpoint regression model

Funding: Health Commission of Hunan Province, China This research was supported by the Health Commission of Hunan Province, China. The funders had no role in study design, data collection and analysis, decision to publish, or preparation of the manuscript.

==============================
Background

In 2011, the United Nations set a target to reduce premature mortality from non-communicable diseases (NCDs) by 25% by 2025. While studies have reported the target in some countries, no studies have been done in China. This study aims to project the ability to reach the target in Hunan Province, China, and establish the priority for future interventions.

Methods

We conducted the study during 2019–2020. From the Global Burden of Disease Study 2016, we extracted death data for Hunan during 1990–2016 for four main NCDs, namely cancer, cardiovascular disease (CVD), chronic respiratory diseases, and diabetes. We generated estimates for 2025 by fitting a linear regression to the premature mortality over the most recent trend identified by a joinpoint regression model. We also estimated excess premature mortality attributable to unfavorable changes over time.

Results

The rate of premature mortality from all NCDs in Hunan will be 19.5% (95% CI [19.0%–20.1%]) by 2025, with the main contributions being from CVD (8.2%, 95% CI [7.9%–8.5%]) and cancer (7.9%, 95% CI [7.8%–8.1%]). Overall, it will be impossible to achieve the target, with a relative reduction of 16.4%. Women may be able to meet the target except with respect to cancer, and men will not except with respect to chronic respiratory diseases. Most of the unfavorable changes have occurred since 2008–2009.

Discussion

More urgent efforts, especially for men, should be exerted in Hunan by integrating population-wide interventions into a stronger health-care system. In the post lock-down COVID-19 era in China, reducing the NCD risk factors can also lower the risk of death from COVID-19.

Introduction

Premature death from noncommunicable diseases (NCDs) remains a major global development challenge in the 21st century. Each year, a total of 15 million people around the world die from NCDs between the ages of 30 and 70 (WHO, 2018). As the most populous country in the world, China is particularly affected by this challenge. A combination of market globalization, rapid urbanization, modifiable risk factors, and population aging over the past decades has led to an NCD epidemic in China: NCDs, mainly including cardiovascular disease (CVD), cancer, diabetes and chronic respiratory diseases, account for 70% of the disease burden and are responsible for 89% of all deaths in the Chinese population (WHO, 2018; Ministry of Finance of the People’s Republic of China, 2020). They have rapidly become the top killer in the country (Li et al., 2017). Meanwhile, the high burden of NCDs reduces effective labor supply and productivity; it also increases treatment costs, thus lowering the accumulation of physical capital and impeding economic growth (Bloom et al., 2018).

In response to the global NCD epidemic, the United Nations (UN) set a target in 2011 for member countries to achieve a relative 25% reduction from the 2010 level in premature mortality from NCDs by 2025 (referred to as the 25 by 25 target)(Kontis et al., 2014). Although studies have estimated the 25 by 25 targets in some countries (Kontis et al., 2014; Malta et al., 2019; Ordunez et al., 2015), no reports for China have been produced. In addition, because the benefits of controlling NCDs produce are realized gradually, it is urgent to ascertain the target feasibility in China to identify essential efforts for future, more effective interventions. Our study therefore projects whether the UN target can be met by 2025 in Hunan Province, China, and when and how much excess premature mortality from NCDs due to unfavorable changes occurred to establish the priority for future, more effective interventions. Additionally, we hope that our research provides a useful reference for countries or regions that are also working to better reduce the risk of premature deaths from NCDs.

Materials and Methods

Data source

We conducted the projection in Hunan Province, Central China, where both the burden of years of lost (YLLs) and the ratio of observed to expected disability-adjusted life-years (DALYs) are significantly higher than the national average (Zhou et al., 2019). Based on the Global Burden of Disease Study (GBD) 2016 for China, we extracted death data from 1990 to 2016 for the above four main NCDs for Hunan by age, sex, year, and underlying causes. The GBD study was a collaboration between the Institute for Health Metrics and Evaluation, University of Washington, and the Chinese Center for Disease Control and Prevention (CDC). With a highly standardized method, the mortality was estimated for China based on multisource death surveillance systems or surveys conducted in the country, which mainly consisted of national disease surveillance points system, population death information registry and management system, national maternal and child health surveillance system, local cancer registry and some other mortality reports (Zeng et al., 2017).

According to the International Classification of Diseases tenth revision (ICD-10), we distributed the ICD-10 codes for the four NCDs as follows: cancer: C00-C97; CVD: I00-I99; chronic respiratory diseases: J30-J98; and diabetes: E10-E14.

Statistical analysis

We set premature mortality from NCDs (also written as premature NCD mortality) as the key indicator for the analysis, with age-standardized rates (ASRs) standardized to the 2010 China census population as a minor indicator. Premature mortality is defined by the WHO as the probability of dying between the age of 30 and 70 from NCDs and is calculated by age-specific death rates with a life table method in the following manner: (World Health Organization, 2020a).

(1) Age-specific death rates in 5-year age groups (e.g., 30–34...65–69) were calculated by 5∗Mx=TotaldeathsfromfourNCDsagedx,x+5Mid−yearpopulationagedx,x+5.

(2) The 5∗Mx was translated into age-specific probability of death:5∗qx=5∗Mx∗51+5∗Mx∗2.5.

(3) The probability of death for persons aged 30–70 was calculated last: 40∗q30=1−∏1−5∗qx

We performed a joinpoint regression to examine trends in premature NCD mortality, with a maximum of three joinpoints set for the analysis. The joinpoint regression describes continuous changes by connecting several line segments on a log scale and identifies statistically significant changes with a Monte Carlo permutation test (Jones et al., 2016; Kim et al., 2000). The three following indicators, annual percentage change (APC), average APC (AAPC) and overall percent change, reflected different changes in temporal trends. The former two indicators were calculated by APCi= {exp(bi) - 1}×100, and AAPC=exp∑wibi∑wi−1×100,

where bi represents the slope coefficients for each segment in the years studied, and wi represents the length of each segment in the intervals. The final indicator was calculated with an AAPC-based exponentiation function by first converting the AAPC to the predicted single-year change and then exponentiating to the number of study years minus one to produce the overall change and its magnitude, which was finally converted to a percent change (Jones et al., 2016).

We projected premature mortality from NCDs (as well as ASRs) with 95% confidence intervals (CIs) for 2025 by fitting a linear regression over the most recent trend identified by the joinpoint model. To estimate the ability to meet the UN target in Hunan, we compared the projected premature mortality from NCDs in the province in 2025 with the level in 2010 to find the relative reduction with the following formula:

relativereduction%=projectedprematuremortality2025−observedprematuremortality2010observedprematuremortality2010∗100.

This allowed us to determine if the relative reduction would be greater than 25%.

To identify excess premature NCD mortality due to unfavorable changes (slowed, stalled or reversed) (Yang et al., 2017), we performed a three-step estimation: first, each most significant APC for the four NCDs was selected as a projection point to find the expected premature mortality, assuming it would continue to decline to 2025 at the same level as the selected APCs. Second, we compared the total differences among observed (1990–2016)-projected (2017–2025) premature mortality with the expected ones to obtain the absolute excess premature mortality. Third, the differences were divided by expected premature mortality to obtain the relative change in the excess premature mortality.

Experimental verification was carried out to evaluate the prediction accuracy of the Joinpoint model. Death data of all NCDs combined during 1990–2011 from the GBD was selected as a sample, to project premature mortality rate for 2012–2016. The results were then compared with the real data with three metrics: mean Square Error (MSE), Percentage Error (PE) and Mean Absolute Percentage Error (MAPE ) (Zeng, 2019). Among them, MSE=1n∑i=1ny ˆ−yi2,PE=y ˆ−yiyi×100%, MAPE=1n∑i=1ny ˆ−yiyi×100%, where y ˆ represents projected value, yi represents observed value. MAPE less than 10% were considered good accurate (Cesnaite et al., 2020).

Analyses for temporal trends and projection through 2025 were performed using the Joinpoint program version 4.7.0.0 (Statistical Research and Applications Branch, National Cancer Institute, USA). Line art was produced by R version 3.6.0 (R Foundation for Statistical Computing, Vienna, Austria). A p-value <0.05 was considered statistically significant.

Results

Temporal trends during 1990-2016 are shown as the AAPC, APC and overall percent change (Table S1). Premature mortality from all NCDs combined was projected to be 19.5% (95% CI [19.0%–20.1%]). The top contributor to premature mortality was CVD (8.2%, 95% CI [7.9%–8.5]%), followed by cancer (7.9%, 95% CI [7.8%–8.1%]). The premature mortality rates for chronic respiratory diseases and diabetes were 1.2% (95% CI [1.2%–1.3%]) and 0.6% (95% CI [0.5%–0.6%]), respectively. Except for a narrow difference in diabetes, men had greater premature mortality from NCDs than women, with an approximately two-fold difference. Regarding the ASRs in 2025, there will be 377.7 deaths (95% CI [367.5–387.8]) per 100,000 persons for all NCDs, with the main contributors being cancer (ASR: 152.5 deaths per 100,000 persons) and CVD (ASR: 143.7 deaths per 100,000 persons) (Table 1).

Table 1 Observed premature mortality from NCDs, ASRs in 2016 and predicted values for 2025, Hunan Province, China.

Sex	Diseases	Observed 2016	Predicted 2025	Percent change in premature mortality rate	
		premature mortality ratea	ASRb	premature mortality rate (95% CI)	ASR (95% CI)		
Both	Total	21.5	428.8	19.5(19.0–20.1)	377.7(367.5–387.8)	−9.3	
	Cancer	8.5	166.6	7.9(7.8–8.1)	152.5(150.1–154.9)	−7.1	
	CVD	9.4	167.6	8.2(7.9–8.5)	143.7(139.4–147.9)	−12.8	
	Diabetes	0.5	8.6	0.6(0.5–0.6)	7.9(7.5–8.3)	20.0	
	Chronic respiratory disease	1.8	27.7	1.2(1.2–1.3)	20.7(19.4–21.8)	−33.3	
	Other NCDs	3.1	58.3	2.9(2.8-3)	52.8(51.9–53.8)	−6.5	
Male	Total	27.5	567.9	26.5(25.8–27.2)	531.4(518.7–544.1)	−3.6	
	Cancer	11.3	219.8	9.7(9.3–10.2)	210.1(206.3–213.9)	−14.2	
	CVD	12.3	223.4	11.8(11.3–12.3)	209.7(202.2–217.4)	−4.1	
	Diabetes	0.5	8.6	0.6(0.5–0.6)	9.2(8.9–9.6)	20.0	
	Chronic respiratory disease	2.5	39.6	2.1(1.9–2.3)	30.9(29.7–32.3)	−16.0	
	Other NCDs	3.9	76.6	3(2.9–3.2)	71.8(70.5–73.1)	−23.1	
Female	Total	14.7	281.1	11.7(11.3–12.1)	222.2(217.3–226.9)	−20.4	
	Cancer	5.6	110.2	4.8(4.7–4.9)	94.2(92.6–95.8)	−14.3	
	CVD	6.3	108.3	4.8(4.5–5.0)	80.6(77.3–84.0)	−23.8	
	Diabetes	0.5	8.5	0.4(0.4–0.5)	6.8(6.5–7.0)	−20.0	
	Chronic respiratory disease	0.9	14.9	0.6(0.4–0.7)	8.5(7.7–9.4)	−33.3	
	Other NCDs	2.2	39.1	1.9(1.8–1.9)	32.5(32.1–33.0)	−13.6	
Notes.

Abbreviations NCDs non-communicable diseases

ASRs age-standardized rates

CI Confidential Interval

CVD cardiovascular disease

a Premature mortality was defined as the probability (%) of dying aged 30–70 from NCDs.

b Rates standardized to the 2010 China census population with age groups 30–34, 35-39... and 65–79 years, in per 100000 populations.

Figure 1 presents the temporal trends for NCDs and the ability to reach the 25 by 25 target in Hunan. A similar trend between the premature mortality and ASRs can be observed. With all NCDs combined, it is not possible to achieve the UN target, as the relative reduction is 16.4%. Among the subcategories, cancer is the least likely to reach the target, with the smallest relative reduction (11.8%). Another disease failing to meet the target would be CVD, with a 22.1% relative reduction. Both chronic respiratory diseases and diabetes shared a more than 25% relative reduction in premature mortality, with the former showing the greater reduction, at 44.0%.

Figure 1 Premature mortality from NCDs and their ASRs in Hunan, China, from observed years (1990-2016) to projected years (2017–2025).

The ability to meet the 25 by 25 target differed across the total NCDs and subcategories: (A) to (C) and (F) will not meet the target with reductions less than 25%, while (D) and (E) will with reductions greater than 25%.

A difference was seen in the distribution of the top two NCDs for both sexes (Fig. 2). In men, CVD remained the top contributor to premature mortality over time, followed by cancer. In women, cancer will take this position in 2025 due to a faster decline in CVD than in cancer. Another difference was the ability to reach the target: a relative reduction of 31.6% in women but only 7.8% in men was projected. In women, except for a slightly smaller reduction for cancer (23.0%), the three other subcategories all showed a greater than 25% reduction. The situation is grim, however, for men, in whom only chronic respiratory diseases achieved a greater than 25% reduction (29.5%), and the result for diabetes even showed a 15.8% increase.

Figure 2 Observed and projected premature NCD mortality by sex in Hunan, China, 1990–2025.

The black dashed line is used to distinguish observed years (1990–2016) from projected years (2017–2025). Five colored lines matching each chart area are used to test whether the target will be met (wider than the chart area in 2025) or not (narrower than the chart area). The ability to meet the 25 by 25 target differed by sex: (A) will not meet the target except with respect to chronic respiratory disease, while (B) will except with respect to cancer.

During 1990–2025, a total absolute excess premature NCD mortality rate of 55.4% and relative excess change of 19.4% were estimated (Fig. 3A). These unfavorable changes mostly occurred from 2008–2009. Among the subcategories (Figs. 3B to 3D), CVD showed both higher absolute excess premature mortality (29.6%) and relative excess change (22.8%) than cancer (absolute excess premature mortality: 14.8%; relative change: 11.4%). The greatest excess change (47.6%) was estimated for chronic respiratory disease, despite its much lower absolute excess premature mortality. The absolute excess premature mortality from diabetes was estimated at only 0.8%, whereas its relative excess reached 11.8% during the same period.

Figure 3 Excess premature mortality from NCDs in Hunan, China, 1990–2025.

Yellow line: projected premature mortality; grey line: favorable premature mortality; blue area: total absolute excess premature mortality. The black dashed lines represent the years in which excess premature mortality began. Whether for all NCDs combined (A) or among subcategories (B to F), most of the excess premature mortality have occurred since 2008–2009.

The Joinpoint model verification showed that for all NCDs combined projected for 2012-2016 (Table 2), the MSE was estimated to be 0.476. A range of 0.70% to 4.48% was estimated for PE, resulting in a MAPE of 2.79%. Among men, the MSE would be 0.721, and the PE would be from 0.65% to 4.61%, resulting in the MAPE at 2.65%. Among women, the matching values were projected to be: MSE at 0.303, PE ranging from 0.99% to 4.68%, and MAPE at 3.26%. These results indicated a good prediction accuracy for the Joinpoint model.

Table 2 Verification results from Joinpoint model: Based on observed premature mortality from NCDs during 1990–2011 and projected values for 2012–2016.

Sex	Items	2012	2013	2014	2015	2016	
Male	projected dataa	28.19	27.88	27.58	27.28	26.98	
	real datab	28.37	28.62	28.91	28.26	27.51	
	PE(%)	0.65	2.59	4.61	3.47	1.93	
	MAPE(%)	2.65					
	MSE	0.721					
Female	projected data	16.03	15.49	14.96	14.46	13.97	
	real data	16.19	15.75	15.64	15.16	14.66	
	PE(%)	0.99	1.67	4.32	4.64	4.68	
	MAPE(%)	3.26					
	MSE	0.303					
Both	projected data	22.56	22.15	21.74	21.34	20.94	
	real data	22.72	22.66	22.76	22.18	21.53	
	PE(%)	0.70	2.24	4.48	3.79	2.74	
	MAPE(%)	2.79					
	MSE	0.476					
Notes.

a Projected data=premature mortality rate from all all NCDs combined projected for 2012–2016.

b Real data=observed premature mortality rate from all all NCDs during 1990–2011.

Abbreviations MSE Mean Square Error

PE Percentage Error

MAPE Mean Absolute Percentage Error

Discussion

Although previous studies have estimated premature mortality from NCDs, each had a different focus. For instance, in the study by Kontis et al. (2014) the authors highlighted the impacts of achieving the WHO agreed six risk factors (tobacco and alcohol use, salt intake, obesity, raised blood pressure and glucose) targets on reaching the 25 by 25 target. The authors calculated a time-based population impact fraction to identify relative reductions in the premature mortality through reanalyses and meta-analyses of epidemiological studies. In another study by Norheim et al. (2015) the authors proposed a more ambitious goal of avoiding 40% of premature deaths from all causes globally by 2030, beyond the current UN sustainable development goal (reducing premature mortality from NCDs by one-third by 2030). They reviewed the UN-based overall 1970–2010 mortality and WHO-based cause-specific 2000–2010 mortality. They concluded such a target could be achieved by moderately accelerating the current mortality decrease during 2000–2010.

Unlike these two studies, our study estimated the 25 by 25 target’s feasibility at the local rather than the global level. As NCDs account for high proportions of the disease burden and all deaths, as mentioned above, we emphasized premature deaths from NCDs rather than all causes to set future control priorities. Based on a Joinpoint regression model, we derived 1990–2016 mortality for Hunan from the GBD 2016 to project premature mortality from NCDs and the excess situation for 2025, assessing the ability to meet the target here. Through the projection, we found that although premature mortality from NCDs in Hunan has continuously declined since 1990, this decline is insufficient to reach the 25 by 25 target. Especially since 2008–2009, almost all NCDs have experienced unfavorable declines. A possible reason is that the Chinese population has experienced adverse changes in both diet and lifestyle over the past decades. According to the national NCD Risk Factor Surveillances Reports, (National Center for Chronic and Non-communicable Disease Control and Prevention, 2012; National Center for Chronic and Non-communicable Disease Control and Prevention, 2016) an increased prevalence of unhealthy diets and physical inactivity has been seen in China. These factors may contribute to the high of 19 ⋅ 5% for premature mortality from total NCDs by 2025, with the major contributors CVD (8.2%) and cancer (7.9%). Considering the baseline in 2010, although it is highly likely that chronic respiratory disease and diabetes will achieve the target, it will be very difficult for CVD and cancer, causing the total NCD to also be unlikely to meet the target. These results indicate that premature NCD deaths remain an urgent heath challenge in Hunan and across the country and that both cancer and CVD are the priority NCDs that need to be immediately addressed.

We also found that men had much higher premature mortality than women, with only chronic respiratory diseases expected to reach the target. One reason for the substantial gender differences is men’s higher prevalence of major NCDs: the prevalence of obesity among men rose more significantly than among women in the decade 2004–2013, rising from 6.1% to 14.0% versus 7.9% to 14.1% among women. Men had a higher prevalence of hypertension and diabetes, but lower performance in awareness, treatment, or management of both diseases (National Center for Chronic and Non-communicable Disease Control and Prevention, 2016; National Center for Chronic and Non-communicable Disease Control and Prevention, 2010). In addition, men are more likely than women to be exposed to key risk factors for NCDsin China, also contributing to the difference (Li et al., 2017). For example, the prevalence of current smoking, drinking (over the past year), and physical inactivity was 51.8%, 58.3%, and 18.2%, respectively in men, compared with 2.3%,15.4%, and 14.3% in women. It is therefore necessary to implement interventions targeting men to tangibly reduce the number of premature NCD deaths.

Most premature NCD deaths can be prevented or delayed by addressing global health risks. Among the modifiable risk factors shared by individuals with NCDs, high blood pressure, smoking, a high-salt diet, and ambient particulate matter pollution (PM, mainly PM2.5) exposure are the four leading factors in China (Zhou et al., 2019). Previous studies have shown that premature mortality from NCDs will not show the most favorable decline unless such factors are simultaneously brought under control (Kontis et al., 2014; Beaglehole et al., 2011). Therefore, a multipronged approach is needed to address the above problems; specifically, an integrated strategy combining a population-wide intervention targeting the above factors with a strengthened health care system is urgently needed, because the benefits of reducing NCD risk factors are produced gradually.

Modified high blood pressure control

Every 10 mm Hg reduction in systolic blood pressure significantly lowered the risk of major CVD events (relative risk 0.80, 95% CI [0.77–0.83]), resulting in a 13% reduction in all-cause mortality (Ettehad et al., 2016). However, high blood pressure management, from awareness to treatment or control, is poor in China (Li et al., 2017). A fact is that only 45% of Chinese adults with hypertension were aware of their condition, only 30% were taking anti-hypertensive drugs, and just 7% had achieved normal blood pressure levels (Lu et al., 2017). A comprehensive, multistage strategy is needed that involves a diet low in salt and rich in polyunsaturated fatty acids, adequate physical activity (no less than a metabolic equivalent of 600 min per week), and an improved primary health-care system. Health authorities and professional institutions need to work together to promote an integrated prevention-control-treatment model at the community level leveraging the internet and health information technology. Within such a model, a hypertension outpatient service in community medical institutions and a family doctor contracting service are required to provide individuals with regular hypertension management with respect to screening, essential anti-hypertensive medications, health counseling, follow-up services, etc. It is particularly crucial for implementing community-based hypertension screening, as it could have a significant long term impact on systolic blood pressure at the population level (Chen et al., 2019b).

More ambitious tobacco control measures

Although the smoking rate has fallen in many high-income countries, it is rising rapidly in China, with a prevalence of 50.5% in male adults (Chinese Center for Disease Control and Prevention, 2019). It would impose a high macroeconomic burden of tobacco-attributable NCDs for China: Tobacco-attributable NCDs would cost China 16.7 trillion yuan (US$2.3 trillion in constant 2018 prices) from 2015 to 2030, equivalent to a 0.9% annual tax on aggregate income. Secondhand smoke exposure would contribute to 14% of the burden (Chen et al., 2019a). As the hometown for of the two best-selling brands of cigarettes in China, Hunan has been slow to take measures for tobacco control. Many successful policies for tobacco control in other countries have shown that a 50% reduction in smoking is feasible (Jha & Peto, 2014; Ng et al., 2014). Such feasibility could be achieved by the following actions: first, raise cigarette taxes. Although China implemented a tax linkage in 2015, raising the wholesale and price taxes on cigarettes from 5% to 11% (National Health Commission of the People’s Republic of China (2015), 2015), cigarettes are much more affordable here than in other countries, with most costing 10 CNY (1.4 USD) a package. It is estimated that a 50% price increase in cigarettes due to taxes in China would yield an additional 231 million years of life (Verguet et al., 2015). Therefore, a higher cigarette tax rate should be the first action. Second, regulate smoking with reference to practices in developed cities such as Beijing, Shanghai, and Shenzhen by comprehensively enacting smoking bans in indoor workplaces, indoor public places, and public transportation. Third, enforce strict bans on tobacco advertising, sponsorship, or any other activity that may weaken smoking control. Fourth, reform the design of cigarette packaging. Cigarette packages in China are all beautifully designed due to a deeply rooted smoking culture, and the warning occupies only a small part of the design space. Packaging should be redesigned to feature the warning in text and graphics. Fifth, conduct targeted health education. The public’s awareness is often confined to ”smoking is harmful to health”, and many people do not know exactly how smoking is harmful. Health education could be conducted by combining traditional media with new media (such as the Internet or WeChat) or through health campaigns hosted by professional doctors.

A stepwise reduction in salt intake

High salt consumption is the leading cause of hypertension and is strongly tied to stroke in China (Mente et al., 2018). Chinese people have a daily average salt intake of 12–14 g, (Mente et al., 2018; Du et al., 2014), much greater than the WHO recommendation of <5 g/day. Due to the characteristics of Hunan cuisine, residents’ diets are particularly high in salt. We suggest a shift strategy for salt reduction involving both commercial foods and consumer behaviors. First, reduce salt in commercial or processed foods through an industry-wide shift. The key is gradual salt reduction in small steps. Following the UK’s practice (He, Brinsden & MacGregor, 2014), gradually lower salt targets (such as a 20% decrease) can be set in high-salt categories for the food industry. It is also encouraged that alternatives with the same or better taste be proposed, such as “less salt, more spices” or a “more potassium, less sodium” diet, which have been shown to be helpful in reducing blood pressure and CVD mortality (Peng et al., 2014). Second, shift consumer awareness to action. National salt campaigns can not only raise consumer awareness but also have a remarkable impact on salt intake in the population. In 2017, the Action on Salt China (ASC) program was established with four cluster randomized controlled trial packages (He et al., 2018). Hunan participated in the program at five county-level locations, but with 130 county-level areas in the province, actual participation was low. Thus, we suggest expanding salt-reducing interventions from the ASC to the whole province. The salt industry should also take responsibility, for instance, by developing saltshakers with smaller holes or convenient salt intake calculators to help consumers make essential behavior changes.

Reduce harmful alcohol intake

Health issues attributable to alcohol use, such as CVD and cancer, have been largely underemphasized in China. During the past 30 years, a striking increase has been seen in alcohol use among Chinese men, greater than that in most other countries (Jiang, Room & Hao, 2015), and this trend is forecasted to continue (Manthey et al., 2019). This increase is strongly associated with robust economic development and a deeply rooted alcohol culture. However, the government can play a substantial role in developing alcohol use policies to alter drinking levels. A good example would be Russia, where WHO’s recommended best buys interventions for alcohol use, including taxation, availability restrictions, and bans on marketing, were effectively carried out, leading to remarkable changes in both alcohol use and the burden of alcohol-related disease (Neufeld & Rehm, 2013). For Hunan, similar interventions are also needed and should be broadened through strict restrictions on alcohol advertising on television, legally binding regulations on alcohol sponsorship, and heavy punishments for drunk drivers.

Multisector cooperation to control PM pollution

Although the WHO has not set targets for environmental risk factors, PM pollution should be seriously addressed due to its striking position as the fourth-leading risk factor for death in China (Zhou et al., 2019). PM pollution causes 3⋅3 million premature deaths worldwide each year, with China being the largest contributor (Lelieveld et al., 2015). Meanwhile, PM pollution has a huge macroeconomic impact on NCD in China, where total losses from NCDs associated with air pollution were estimated to be $499 billion (constant 2010 USD) from 2015–2030 (Chen & Bloom, 2019). As in many other regions in China, PM2.5 reduction control in Hunan has just begun. The local government released in 2018 a 3-year action plan to reduce the annual PM2.5 concentration to less than 40 µg/m3 by 2020 (People’s Government of Hunan Province, 2018), but this plan still falls far short of the WHO guidelines (World Health Organization, 2020b). Thus, intersector collaboration in public–private partnerships should be encouraged by setting PM2.5 levels as an assessment indicator for local government development, supervising pollution from industrial enterprises, establishing reward-and-punishment mechanisms, prompting responses to heavy pollution weather, etc.

When discussing NCDs control, we must recognize that the COVID-19 is now capturing the world focus. As the COVID-19 pandemic is raging worldwide and spreading fast in many countries, China has largely controlled the epidemic. The Chinese government has spent substantial efforts in controlling the COVID-19 epidemic—such as lockdown, extension of the lunar new year holiday, and facility isolation of mild to moderate cases using Fangcang shelter hospitals (Chen et al., 2020b; Chen et al., 2020c). However, these measures impose costs, involving human resources, economic losses, public engagement, coordinated governance structures at the national and local levels, etc. Thus we need to think about long-term plans for epidemic control. In the post lock-down COVID-19 era in China, the management of NCDs becomes especially important as people with underlying chronic diseases are more likely to die from COVID-19 (Wu & McGoogan, 2020). Reducing the risk factors in NCDs such as smoking and air pollution can also lower the risk of death for COVID-19 (Chen et al., 2020a; Alqahtani et al., 2020).

Our study is subject to some limitations. First, the data were derived from the GBD 2016, and all the limitations in the GBD study are also applicable to this study. Second, only five risk factors were addressed in the recommended interventions because the effects of other factors can be partially replaced, but this may affect the most effective control of future premature NCD deaths. Third, we conducted the projection under the current trend, with no consideration for possible greater efforts to reduce NCD risk factors in the future. However, the benefits of lowering the risk factors are produced gradually, which should have a small impact on the results of the present study. Additionally, our projection did not consider the change of age structure of the population during the period studied. This may weaken to some extent, the extrapolation of the model presentation.

Conclusions

Despite a continuous decline in premature mortality from NCDs in Hunan, China, the decline slowed ten years ago. Premature NCD deaths remain high and are unlikely, particularly in men, to reach the 25 by 25 target by 2025. More bold actions combining population-wide interventions for key risk factors with improved health-care systems are urgently needed.

Supplemental Information

Supplemental Information 1 Temporal trends in premature mortality from NCDs during 1990-2016, Hunan Province, China

Abbreviations: APC, Annual percent change; AAPC, Average Annual percent change; NCDs, non-communicable diseases; CVD, cardiovascular disease. a It was calculated by first converting AAPC to predicted single year change, then exponentiating to the number of study years minus one to produce overall change and its magnitude, which was finally converted to a percent change.

* Statistically significant at the alpha=0.05 level.

Click here for additional data file.

Supplemental Information 2 Raw measurements

The standardized population, the extracted death data from the GBD study 2016, and the corresponding variables in the death data.

Click here for additional data file.

We expres our thanks to the Institute for Health Metrics and Evaluation, University of Washington and the Chinese CDC, for their collaborative work in producing the GBD results for China. We thank Professor Guoqing Hu from Xiangya School of Public Health, Central South University, China, for his advice on the paper.

Additional Information and Declarations

Competing Interests

Author Contributions

Data Availability

The authors declare there are no competing interests.

Qiaohua Xu conceived and designed the experiments, performed the experiments, analyzed the data, prepared figures and/or tables, authored or reviewed drafts of the paper, and approved the final draft.

Maigeng Zhou conceived and designed the experiments, authored or reviewed drafts of the paper, and approved the final draft.

Donghui Jin and Li Yin performed the experiments, authored or reviewed drafts of the paper, and approved the final draft.

Xinying Zeng, Jinlei Qi, Yuan Liu and Lei Yin analyzed the data, prepared figures and/or tables, and approved the final draft.

Yuelong Huang conceived and designed the experiments, performed the experiments, authored or reviewed drafts of the paper, and approved the final draft.

The following information was supplied regarding data availability:

The raw measurements are available in the Supplementary File.

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
