# Peer review of "Projection of premature mortality from noncommunicable diseases for 2025: a model based study from Hunan Province, China, 1990–2016"

_PeerJ, doi:10.7717/peerj.10298_

## Round 0.1 · original submission · Major Revisions

Excuse for delay in revision due to bias in the reviewers' estimates. This work needs revision. Please check the statistics as suggested by reviewer #4 and comparison to previously published methods as suggested by reviewer #2. Despite some criticisms, I recommend revise and resubmit. Waiting revised version.

Reviewer 1 ·

Basic reporting

no comment

Experimental design

no comment

Validity of the findings

no comment

Additional comments

The present study project the ability to reach the target to reduce premature mortality from NCDs by 25% by 2025 in Hunan Province, China, and establish more urgent efforts, particularly on men health, for future interventions. The study presents the results of original research and statistics and analyses are performed to a high technical standard and are described in sufficient.

Minor comments:
p6. line 44 what are the NCDs on China? What diseases are included on NCD list? clarify on text.

p.7 line 69. clarify what systems or surveys

p. 12 line 192: refference.

p. 15 Study limitations can exist due to constraints on research design or methodology, and these factors may impact the findings of your study. "estimations should be interpreted with caution": this sentence minimizes the importance and reliability of your study. So, remove this topic and dicuss this on first paragraph of discussion.

Reviewer 2 ·

Basic reporting

The paper is interesting and addresses an important topic. Overall, the English writing is professional and clear. I have several comments hoping to further improve the paper.

The authors should add citations to back the claims in several places.

-- line 50: ... by 25 targets in some countries, no reports for China have been produced. Please cite previous studies estimating 25*25 targets.

-- line 80: citation for life table method is wrong.

Experimental design

1. Please use the most up-to-date GBD data. As far as I know, there is already GBD 2017, the authors should double-check and use the latest data source.

2. The health statistical year book (2020) in China could have the most up-to-date death data for Hunan. If so, it would be better to use the health statistical year book rather than GBD data.

3. line 82: better to write the denominator as "mid-year population aged (x, x+5)"

4. better to discuss the differences in methods and findings with previous studies on this topic:

-- Kontis V, Mathers CD, Rehm J, Stevens GA, Shield KD, Bonita R, Riley LM, Poznyak V, Beaglehole R, Ezzati M. Contribution of six risk factors to achieving the 25× 25 non-communicable disease mortality reduction target: a modelling study. The Lancet. 2014 Aug 2;384(9941):427-37.

-- Avoiding 40% of the premature deaths in each country, 2010–30: review of national mortality trends to help quantify the UN Sustainable Development Goal for health

Validity of the findings

The discussions are well-written. I think the paper can be stronger by adding more discussions on 1) the broader social and economic impact of NCDs and related risk factors in China, 2) effective prevention strategies, and 3) the covid-19 pandemic, and 4) the assumptions in this study and how the assumptions may affect the results.


1) The broader social and economic impact of NCDs and related risk factors:

-- Could further expand the discussion of the NCDs on the macroeconomic impact: the high burden of NCDs reduces effective labor supply--both through mortality (reduction in quantity) and morbidity (reduction in quality); it also increases treatment costs, thus lowering the accumulation of physical capital and impede economic growth. Could cite: Bloom DE, Chen S, Kuhn M, McGovern ME, Oxley L, Prettner K. The economic burden of chronic diseases: Estimates and projections for China, Japan, and South Korea. J Econ Ageing 2018; [Epub ahead of print];

-- As for PM pollution, better to add discussions on the total losses from NCDs associated with air pollution in China-- $499 billion (constant 2010 USD) from 2015–2030. Could cite: the Chen S, Bloom DE. The macroeconomic burden of noncommunicable diseases associated with air pollution in China. PLoS One 2019; 14(4): e0215663.

-- Tobacco-attributable NCDs affect China’s productive capacity--impose a total cost of 16.7 trillion yuan (US$2.3 trillion, in constant 2018 prices) in the period 2015–30, which corresponds to an annual tax of 0.9 percent on aggregate income. Secondhand smoke exposure accounts for 14 percent of the burden. Could cite: Chen S, Kuhn M, Prettner K, Bloom DE. Noncommunicable Diseases Attributable To Tobacco Use In China: Macroeconomic Burden And Tobacco Control Policies. Health Affairs 2019.

2) The prevention strategies that can help in reaching the target:

--When discussing the high blood pressure control issues, the author can add that community based hypertension screening and encouraging people with raised blood pressure to seek care and adopt blood pressure lowering behaviour changes could have important long term impact on systolic blood pressure at the population level. Could cite: -- Ettehad D, Emdin CA, Kiran A, et al. Blood pressure lowering for prevention of cardiovascular disease and death: a systematic review and meta-analysis. Lancet 2016;387:957-67. doi:10.1016/S0140- 6736(15)01225-8; Chen S, Sudharsanan N, Huang F, Liu Y, Geldsetzer P, Bärnighausen T. Impact of community based screening for hypertension on blood pressure after two years: Regression discontinuity analysis in a national cohort of older adults in China. BMJ 2019; 366: l4064.; Sudharsanan N, Chen S, Garber M, Bärnighausen T, Geldsetzer P. The effect of home-based hypertension screening on blood pressure change over time in South Africa. Health Affairs 2020; 39(1): 124-32.

-- And the factor that only 45% of Chinese adults with hypertension were aware of their condition, only 30% were taking antihypertensive drugs, and just 7% had achieved normal blood pressure levels. Could cite: Lu J, Lu Y, Wang X, et al. Prevalence, awareness, treatment, and control of hypertension in China: data from 1·7 million adults in a population-based screening study (China PEACE Million Persons Project). Lancet 2017;390:2549-58. doi:10.1016/S0140- 6736(17)32478-9

3) Adding some discussions on the importance of managing chronic diseases during the covid-19 pandemic

China has spent substantial efforts in controlling the epidemic-- such as lockdown, extension of lunar new year holiday, and facility isolation of mild cases using Fangcang shelter hospitals, however, these measures impose costs (can cite:--Chen S, Yang J, Yang W, Wang C, Bärnighausen T. COVID-19 control in China during mass population movements at New Year. The Lancet 2020; 395(10226): 764-6.
--Chen S, Zhang Z, Yang J, et al. Fangcang shelter hospitals: a novel concept for responding to public health emergencies. The Lancet 2020; 395(10232): 1305-14.). Thus we need to think about long-term plans for epidemic control. In the post lock-down covid-19 era in China, management of chronic diseases becomes especially important as people with underlying chronic diseases are more likely to die from covid-19 (cite: Wu Z, McGoogan JM. Characteristics of and important lessons from the coronavirus disease 2019 (COVID-19) outbreak in China: summary of a report of 72 314 cases from the Chinese Center for Disease Control and Prevention. Jama. 2020 Apr 7;323(13):1239-42) and reducing the risk factors in chronic diseases such as smoking and air pollution can also lower the risk of death for covid-19. (could cite: Chen K, Wang M, Huang C, Kinney PL, Anastas PT. Air pollution reduction and mortality benefit during the COVID-19 outbreak in China. The Lancet Planetary Health. 2020 Jun 1;4(6):e210-2.)

4) better to add discussions on the assumptions in this study and how the assumptions may affect the results

Reviewer 3 ·

Basic reporting

The article is in clear and unambiguous, professional English.
The article meets the requirements of the journal.

Experimental design

No comment

Validity of the findings

The article complies with the standards of the journal PeerJ.
On the basis of statistical data, the authors convincingly show a clear trend of increasing mortality in four non-communicable diseases: cancer, cardiovascular disease, chronic respiratory diseases and diabetes. According to these estimates, it is impossible to achieve by 2025 the United Nations projected 25% reduction in the rate of premature mortality from non-communicable diseases by 2025.

Additional comments

The article is of broad and interdisciplinary interest and meets the requirements of the journal.

Reviewer 4 ·

Basic reporting

The presented work is devoted to a very serious problem - the analysis of the dynamics of mortality of the population aged 30-70 years in a large province of China. The authors convincingly show that while maintaining the existing trends, it will not be possible to achieve the stated goal: «to reduce premature mortality from non communicable diseases by 25% by 2025».
The strength of the work is that they did not limit themselves to approximation and extrapolation of data, but used data in age groups. This makes it possible to obtain a more accurate forecast, since the dynamics of changes in the age structure of the population is taken into account.

Experimental design

Not applicable. The article is devoted to the analysis of medical reporting data. The statistical methods used correctly (as far as can be judged, without the initial data).

Validity of the findings

The figures and tables presented are in accordance with the requirements.
Also, there are no comments on the structure of the article.
Literature well referenced and relevant.

Additional comments

1. Raw data not supplied.

2. When determining mortality in 5-year age groups (lines 81 and 82), the terms “age grops”, “Total death ... between age (x) and exact age (x + 5)”, “Total population between age (x ) and exact age (x + 5) ". However, in medical reporting age is usually given as “full years,” as a result, the group “55 to 60 years” would include 6 years instead of 5. The term "exact age" used in the definition is not unambiguous. Therefore, it is desirable to give an example of the definition of age groups.
A more unambiguous definition of the concept of “age” used in the article is also required, since age can be determined both by the year of birth and at the time of the origin of the event under study. In this case, the population is usually determined as at the beginning of the year, and for death - the age at the time of death.
If the age in determining the population size and in the deceased is determined differently, this leads to a systematic error in determining the dependence of mortality on age, and it is advisable to make adjustments for this bias effect (by an average of 0.5 years).

3. When predicting the dynamics of mortality, the authors use statistical methods that not only give a forecast, but also estimate its expected accuracy. However, these methods evaluate the accuracy of the forecast using a number of assumptions that may not be met. In such cases, it is advisable not to restrict ourselves to the obtained estimate of the forecast accuracy, but also to carry out experimental verification.
In this regard, I would recommend the authors (if technically feasible) to carry out such a check and provide data on its results in the article.
In this regard, I would recommend the authors (if technically feasible) to carry out such a check and provide data on its results in the article.
This can be done, for example, in the following way. The work uses data for 1970-2016. It is possible to forecast the data for 1970-2011 and compare the results with the actual data for 2012-2016.

---

## Round 0.2 · accepted · Accept

Thank you for the update and detailed answers. All the reviewers' remarks were taken into account. We may accept the manuscript now.

Reviewer 2 ·

Basic reporting

It's a nice contribution to the literature.

Experimental design

no comment

Validity of the findings

no comment

Reviewer 4 ·

Basic reporting

.

Experimental design

.

Validity of the findings

.

Additional comments

In my review of the initial version of the article, several wishes were expressed. All of them were fully taken into account by the authors in the revised version of the article.